# Current Status of the Instructional Cues Provided by Notochordal Cells in Novel Disc Repair Strategies

**DOI:** 10.3390/ijms23010427

**Published:** 2021-12-31

**Authors:** Ajay Matta, William Mark Erwin

**Affiliations:** 1Notogen Inc., Toronto, ON M5G OB7, Canada; amatta@notogen.com; 2Department of Surgery, University of Toronto, Toronto, ON M5T 1P5, Canada

**Keywords:** notochordal cells, spinal pain, regenerative medicine, degenerative disc disease

## Abstract

Numerous publications over the past 22 years, beginning with a seminal paper by Aguiar et al., have demonstrated the ability of notochordal cell-secreted factors to confer anabolic effects upon intervertebral disc (IVD) cells. Since this seminal paper, other scientific publications have demonstrated that notochordal cells secrete soluble factors that can induce anti-inflammatory, pro-anabolic and anti-cell death effects upon IVD nucleus pulposus (NP) cells in vitro and in vivo, direct human bone marrow-derived mesenchymal stem cells toward an IVD NP-like phenotype and repel neurite ingrowth. More recently these factors have been characterized, identified, and used therapeutically to induce repair upon injured IVDs in small and large pre-clinical animal models. Further, notochordal cell-rich IVD NPs maintain a stable, healthy extracellular matrix whereas notochordal cell-deficient IVDs result in a biomechanically and extracellular matrix defective phenotype. Collectively this accumulating body of evidence indicates that the notochordal cell, the cellular originator of the intervertebral disc holds vital instructional cues to establish, maintain and possibly regenerate the intervertebral disc.

## 1. Introduction

A host of biological factors have been postulated to offer restorative effects upon degenerative intervertebral disc (IVD) including growth factors (including a variety of members of the transforming growth factor family—such as bone morphogenic proteins ‘BMPs’) as well as anti-catabolic factors such as inhibitors of nuclear transcription factors such as Nuclear Factor kappa-light-chain-enhancer of activated B cells (Nfκβ), and Wnt-β-catenin. Many of these proteins have been studied based upon developmental biology or have been repurposed from other applications as in the case of certain BMP family members (BMP-7).

However, nature has provided a unique set of circumstances wherein certain species, such as rabbits, pigs and outbred or ‘non-chondrodystrophic’ (NCD) canines, retain primitive, developmentally important notochordal cells through skeletal maturity and the discs of these animals are protected from degenerative disease [1,2]. In contrast, notochordal cells disappear early in life for both humans and chondrodystrophic (CD) canines, heralding the onset of degenerative changes on the part of the notochordal cell-poor human and CD canine IVDs. Thus, there exists a naturally occurring dichotomy within the sub-species of canine where the most salient difference is the persistence of notochordal cells in the NCD canines. It has been confirmed by several groups that conditioned media obtained from notochordal cell tissue culture confers anabolic, anti-apoptotic, and anti-catabolic effects on cells derived from the IVD NP thus validating that such factors confer instructive cues to the heterogenous cell population within the IVD [1,3,4,5,6,7,8,9]. Others have suggested that the extracellular matrix of the notochordal cell rich IVD NP itself may provide instructional cues that could aid in disc repair [10]. With respect to the best approach to address IVD repair, the notochordal cell and associated in vivo milieu has attracted considerable interest over the past years. Recent publications have shed new light on the instructional clues provided by notochordal cells with respect to homeostatic regulation of the IVD and how these soluble cues could be harnessed as a molecular therapy.

## 2. Anatomical/Physiological Role Played by the Notochord in Disc Development and Homeostatic Regulation of the IVD

During the initial stages of development, the notochord provides structural integrity, defines the longitudinal axis of the developing embryo, and provides instructional cues to induce local undifferentiated mesenchymal cells to migrate, condense, and differentiate; all of which are central to the formation of the vertebral column and the developing spinal cord [11,12,13,14,15]. During vertebrogenesis, the notochord ultimately undergoes segmentation during and forms the centre of the intervertebral disc—the nucleus pulposus under the influence of the T-box transcription factor-commonly referred to as “T” (brachyury) along with Sonic Hedgehog (SHH), Noggin and Pax1 [15,16,17,18].

Notochordal cells can persist within the IVD into adulthood in some animal species with this phenomenon thought to occur at least in part due to ongoing activity of the transcription factors Sox5 and Sox6, however the precise mechanisms remain to be determined [13,19]. Loss of function in in vivo models have validated the impact of these transcription factors where *Sox*5−/− and 6−/− mutant mice display a failure to form a nucleus pulposus, along with a severely distorted vertebral column confirming the pivotal role that these genes play in directing the development, maturation, and survival of the disc nucleus pulposus [13]. Amongst the proteins involved with the development of the IVD, connective tissue growth factor (CTGF/CCN-2) is an important notochordal cell-secreted matricellular growth factor-like protein that plays an early role in development and in the formation of the notochord that involves inhibition of canonical signaling (involving β-catenin) as well as non-canonical signaling including interaction with the WNT receptor complex [20]. Studies involving deletion of CCN2/CTGF genes in mice have shown disrupted IVD formation in embryonic and newborn mice with impaired expression of type II collagen and aggrecan and premature degeneration marked by the development of osteophytes and disc herniations [21]. During embryogenesis matrix development is under TGF-β direction that also involves sclerotome and annulus fibrosus cell differentiation [11,18,22]. It has been reported that Smad3 knock out mice develop a severe malformation of the spine involving kyphosis and pronounced, early degeneration, validating the important role of downstream Transforming Growth Factor β (TGF-β) signaling within the Smad cascade [23]. Furthermore, in the developing mouse spine (virtually purely notochordal in composition) it has been shown that there is a significant increase in TGF-β1 expression at P0 as compared with E12.5 mice, emphasizing the importance of TGF-β signaling in development [23]. In addition, knock down of TGF-β receptor 2 (TGF-βrII) in mice leads to reductions in downstream TGF-β Smad-related signaling and accelerated degenerative disc changes including reduced cartilage endplate tissue, increases in MMP-13 expression and related ECM degradation [24]. These findings are important and highly relevant to the question of TGF-β and Smads with respect to notochordal cells since the developing IVD NP is highly notochordal in content-particularly in rodents. In another study porcine IVD NPs (almost purely notochordal in nature) were freeze dried, pulverized, and lyophilized and used to treat human IVD NP cells in vitro [25]. In this study it was reported that notochordal-derived matrix (NDM) induced robust Smad signaling in canine and human IVD NP cells strongly supporting notochordal cell-related pro-Smad activation [25].

It has further been demonstrated that TGF-β receptor 2 proteins (TGF-βr2) are intimately involved with the development and maintenance of the normal IVD phenotype such that deletion of the TGF-βr2 gene led to abnormal development of the IVD [24]. Other work involving Smad3 gene deletion experiments has resulted in kyphotic malformation of the IVD with aberrant degeneration including reduced disc height, and decreased collagen and proteoglycan production [23]. With respect to Smad3 activity, gain and loss of function experiments have shown that Smad3 (at least in part in concert with AP-1) can modulate the activity of the CTGF promoter [26]. This interaction within the canonical TGF-β signaling pathway provides further evidence that TGF-β can induce CTGF expression via activation of Smad3 [26].

TGF-β receptor signaling typically occurs downstream of TGF-β1 binding to the TGF-β receptor 1 (r1) that in turn dimerizes with TGF-β receptor 2 (r2). Upon phosphorylation TGF-βr2 induces phosphorylation of Smad 2/3 forming a heterotrimer involving Smad4 that leads to translocation of the complex to the nucleus and transcription of TGF-β-related genes [27]. In cre-inducible mouse models, Jin et al. have shown that TGF-β receptor signaling is vital for growth and development of the disc during post-natal stages and that dominant negative phenotypes develop significantly elevated mRNA levels of Col 10a1, MMP-13, ADAMTS4, and ADAMTS5 demonstrating the pro-catabolic effect of reduced TGF-β signaling [24]. As described above, TGF-β activation and Smad signaling induces transcription of CTGF/CCN2 that leads to feedforward induction of both CTGF/CCN2 and TGF-β1, thus providing further evidence of the importance of these two growth factors in IVD development, maintenance and attempts at repair under pro-degenerative conditions [28]. It is interesting to note that some reports have shown an increase in TGF-β1 expression in degenerative discs whereas others have shown this growth factor to be of low or undetectable amounts in surgical samples [7,29]. These observations when taken together provide evidence for an incompletely understood pleiotropic mechanism involving growth factors, cytokines and complex cell-extracellular matrix regulation within the IVD [7,26].

## 3. Notochordal Cell Associated Extracellular Matrix Interaction

Like many other tissues, function of the IVD NP is dependent upon cellular–ECM interaction, principally involving proteoglycans such as aggrecan, notable for its role within the IVD as it is the major proteoglycan responsible for water binding and loadbearing [30,31,32,33]. Another class of proteoglycans that is integral to cellular–ECM interaction and ECM maintenance known as small leucine-rich proteoglycans (SLRPs) play roles in the organization of collagen, growth factor sequestration and effects upon inflammation and function of the ECM [34]. The contribution and influence of these SLRPs to ECM maintenance continues to evolve, however degeneration of these SLRPs compromises their function and has negative consequences upon the ECM [34]. Decorin for example, plays a vital role in the maintenance of the ECM with a deficiency in Decorin associated with aberrant collagen content, apoptosis, and the development of fibrosis. Rajasekaran et al. in a study of normal vs. degenerative human IVD tissues, showed a correlation between reduced collagen type 2 and SLRPs within the nucleus pulposus [35]. These findings are important in that SLRPs are integral to growth factor binding/presentation to cells and robustly influence the cross-linking of collagen, underscoring their importance in ECM maintenance and metabolism [35]. A manuscript published by Erwin et al. used iTRAQ proteomic methods, Western blotting and quantitative biomechanical methods and reported that the core proteins within SLRPs obtained from notochordal cell-rich non-chondrodystrophic canines were intact whereas non-chondrodystrophic canine IVD NPs yielded fragmented SLRP core proteins [36]. These observations indicate that the SLRPs within chondrodystrophic canines were already undergoing natural degeneration and were not firmly bound to the matrix. These observations when coupled with inferior biomechanical properties on the part of the notochordal cell-poor chondrodystrophic canines indicate that the presence of notochordal cells results in an IVD NP that resists natural degeneration and maintains biomechanical properties. Taken together, this study lent credence to the hypothesis that notochordal cell rich IVD NPs provide for a healthy IVD NP as compared with IVD NPs that are naturally notochordal cell deficient.

## 4. Notochordal Cell Secreted Proteins

In animals protected from developing DDD (such as rabbits, pigs, non-chondrodystrophic canines), notochordal cells exist within a unique milieu that appears to resist catabolic/pro-inflammatory events central to the progression of DDD [1,3,4,37]. The reasons for the disparity in susceptibility to DDD in notochordal-rich IVDs as compared with notochordal cell-deficient IVDs has been the subject of much research interest [1,37,38]. Theoretical reasons for the protective role played by notochordal cells have included possible biophysical properties of these cells as well as a role played by notochordal cell-secreted soluble factors. Numerous in vitro studies using growth media conditioned by notochordal cells and/or notochordal cell-rich IVD NP tissues have demonstrated an anabolic, anti-catabolic and anti-apoptotic effect of such conditioned media upon IVD NP cells thus supporting a role for such soluble as a putative treatment for DDD [1,3,6,39,40]. More recently in vivo pre-clinical animal studies have conclusively demonstrated that notochordal cells secrete soluble factors that inhibit IVD cell apoptosis, suppress degeneration and ingrowth of neurites, and can induce a reparative effect [4,5,6,8,41,42]. A recent comprehensive secretome analysis of the notochordal cell-rich non-chondrodystrophic canine IVD NP demonstrated that amongst the thousands of secreted proteins, connective tissue growth factor (CTGF) and transforming growth factor beta 1 (TGF-β1) figured prominently and when used in combination in the form of an injectable agent, recapitulate the impressive anti-degenerative/pro-repair effects of a single injection of notochordal cell conditioned medium (NCCM) [7]. A follow-up study demonstrated the efficacy of a single injection of rhCTGF + rhTGF-β1 (within an excipient solution, “NTG-101”) into the rat caudal discs and a large animal model of DDD (CD canine needle puncture injured discs) [8]. Within this study, it was demonstrated that the injection of NTG-101 significantly reduced the expression of pro-inflammatory cytokines, and enzymes, stimulated the production of aggrecan and collagen type 2, and maintained disc height and biomechanical properties compared with injured and vehicle injected discs in a biomechanically relevant large animal model of DDD [8]. This study provides important evidence in support of the role played by notochordal cell-secreted soluble factors (specifically CTGF and TGF-β1) as therapeutic proteins with which to treat DDD.

## 5. Influence of Notochordal Cells upon Mesenchymal Stem Cells

The intervertebral disc has been shown to host a robust population of stem cells of mesenchymal lineage [43,44]. Stem cells are known to play vital roles in repair and the otherwise dormant ‘niche’ containing stem cells can be activated in time of injury/disease [45]. In the event of trauma, tissue healing is dependent upon the degree of injury as well as the intrinsic capacity of the injured tissues to recover. Central to this ability to heal is the ability of inflammation, the immune system and otherwise dormant stem cells to activate tissue repair that includes mobilizing stem cells to migrate from their quiescent niche towards the injury site and differentiate into the appropriate cellular phenotype required for healing [45]. Studies have shown that stem cell populations within the IVD undergo a kind of ‘fatigue’, likely associated with chronic pro-inflammatory cytokine activity, and limited nutrition [44].

Interestingly, it has been shown that notochordal cell-conditioned medium can induce mesenchymal stem cells to differentiate towards an IVD NP-like phenotype further supporting the pro-anabolic/restorative effects of soluble factors produced by notochordal cells. Korecki et al. [46], studied the ability of porcine-derived notochordal cell-conditioned medium (NCCM) compared with ‘chondrogenic medium’ (containing Dexamethasone + TGF-β3) to induce change in phenotype from bone marrow-derived stem cells to one resembling IVD cells. Korecki et al. hypothesized that notochordal cell-secreted soluble factors and/or notochordal cell matrix supply ‘instructional cues’ that may influence MSCs differentiation. These authors reported that NCCM induced MSC differentiation into a more ‘nucleus pulposus-like cell’ than did ‘chondrogenic medium’, however a definitive mechanism remains elusive [46]. Another study by Li et al. [47] reported that MSCs may differentiate into a notochordal cell-like phenotype under the influence of NCCM and cite previous reports that vital instructional cues to promote such differentiation may be TGF-β1 and CTGF as has been previously reported [7]. Both cited studies indicate that understanding the cellular signaling mechanisms involved with stem cell differentiation into an IVD NP phenotype is a necessary goal, however these pathways have yet to be determined.

The results of these studies lend further support to the hypothesis that notochordal cell-secreted factors confer a pro-anabolic/reparative effect upon a host of cells including IVD chondrocyte-like cells within the nucleus pulposus as well as mesenchymal stem cells [46].

## 6. Notochordal Cell-Rich Extracellular Matrix Instructional Cues

As described above, tissue culture media conditioned by notochordal cell-rich IVD NP tissue (NCCM) has been shown to confer anti-apoptotic, pro-anabolic and anti-catabolic effects upon IVD NP cells (in vitro and in vivo) and upon articular chondrocytes in vitro [3,4,5,6,7,10,48]. Several other studies have postulated that acellular, IVD extracellular matrix may confer beneficial effects upon IVD NP cells. These studies used lyophilized notochordal cell-rich IVD NP tissue that was then suspended within growth medium and then used to culture bovine NP cells [10]. Freeze drying, pulverizing, and lyophilizing the notochordal cell-rich IVD would necessarily include the notochordal cells themselves, their secreted products and other ECM proteins associated with the notochordal cell-rich IVD NP. In the published work by de Vries et al., in vitro methods were used to determine that notochordal cell matrix (NCM) was capable of stimulating bovine caudal disc NP cells to stimulate the production of collagen type II as well as glycosaminoglycans and proteoglycans such as aggrecan [10]. These same authors also reported that NCM induced a stronger anabolic/proliferation effect upon bovine NPCs than did NCCM. However, there was no specific mechanism hypothesized. Meanwhile, Matta et al. [49] reported that the activity of CTGF and TGF-β1 inhibit downstream phosphorylation of p38 Map Kinase and the p50/p65 subunits of the NFκβ signaling pathways [49]. Inhibition of these pro-inflammatory pathways suppresses ECM degradation. Additionally, these two notochordal cell-secreted growth factors elevate ERK1/2 and AKT phosphorylation via increased Smad signaling TGF-β1 [49]. Furthermore, integrin-mediated signaling from such as via CTGF in addition to increased Smad signaling induces pro-survival and ECM synthesis [49]. Presumably NCM would induce similar pathways since such a conditioned matrix would necessarily include at least these factors (and possibly others), however any additional signaling associated with unknown components of NCM remains unknown. Since the studies concerning ‘NCM’ did not determine any anabolic effects attributed to the ECM proteins within ‘NCM’ as distinct from the notochordal cells themselves (and their secreted factors) it is difficult to conclude what contribution the actual matrix within NCM may have provided, since notochordal secreted and produced factors would have undoubtedly contributed to any biological effect. Finally, NCM is a complexly uncharacterized formulation likely containing thousands of proteins and as such, the translational capacity of such an approach is difficult to understand. Nonetheless, this report further captures the importance of notochordal cells to homeostatic regulation of the IVD and potential for disc repair strategies.

## 7. Influence of Notochordal Secreted Factors upon Neural Ingrowth into the Degenerative Disc

It is well known that the nucleus pulposus of the healthy IVD is aneural with nociceptive and mechanoreceptive innervation confined to the outer few millimeters of the annulus fibrosus [50]. Innervation within the annulus fibrosus principally provides nociception as well as instructional clues to spinal musculature to control motor function and coordinate upright posture. However, temporally associated with the development of fissures and tears within the degenerating disc, is ingrowth of nociceptive capable neurons that penetrate the inner nucleus pulposus, thus providing intradiscal nociception and facilitating the generation of discogenic pain [50,51,52,53]. DDD is associated with inflammatory cytokine-driven losses of collagen type 2 and proteoglycans (principally aggrecan) that summate to diminish the biochemical and biomechanical properties of the IVD. Interestingly, the healthy IVD NP that contains substantial proteoglycans such as aggrecan have been shown to repel the ingrowth of neurons into deeper levels of the AF as well as the NP [54]. Also, conditioned media obtained from notochordal-rich IVDs inhibited neurite outgrowth from a multitude of neural cell types in vitro while maintaining cell viability [42]. In fact, only when the glycosaminoglycans were digested was this neurite inhibition effect reversed, leading to the hypothesis that intact GAGs (notably chondroitin sulfate) synthesized by notochordal cells within healthy IVDs contribute to the lack of neural ingrowth into the IVD. It is a tantalizing notion that notochordal cell-secreted factors not only serve as anti-inflammatory, anti-catabolic, and pro-anabolic mediators, but that they may also serve to inhibit the development of nociceptive elements within the IVD.

The primary rationale for molecular therapy to address DDD is to influence the pain and disability associated with this condition. With respect to possible influences of notochordal cell-secreted factors and pain, a recent “Best Paper” abstract presentation at the 2020 North American Society Annual meeting addressed this issue. In this abstract, Matta et al. used tissue sections obtained from beagle IVDs that were previously needle puncture injured and then four-weeks later, injected with either saline or NTG-101 and harvested 14-weeks later [8]. These tissue sections were obtained from identical IVDs that, after NTG-101 injection, preserved disc height and demonstrated improved biomechanical properties and robust anti-catabolic/pro-anabolic effects, whereas saline injected discs lost disc height, had inferior biomechanical properties and underwent significant pro-catabolic changes [8]. Matta et al. showed that the NTG-101 injection also inhibited the expression of the neurotrophins nerve growth factor receptor (TrkA), brain-derived neurotrophic factor (BDNF), and the BDNF receptor (TrkB) as well as the neuropeptide calcitonin gene related peptide receptor (CALCRL). These neurotrophins and neuropeptides are known to be highly expressed in patients exhibiting painful DDD [51,52,55,56]. The suppression of neurotrophin and neuropeptide expression in IVDs injected with NTG-101 but not saline supports the hypothesis that molecular therapy based upon the notochordal cell secretome (NTG-101) may be effective at decreasing IVD generated pain [57].

## 8. Dichotomy of IVD NP Cell Secreted Growth Factors upon IVD Health and Disease

Given the reported benefits of intradiscal-injection of rhCTGF + rhTGF-β1 in rodent and large animal models of DDD, it is interesting to note that various groups have reported both elevated and suppressed/absent levels of TGF-β1 in degenerative IVDs [23,29]. At least some of these variable observations may have to do with species-specific differences such as aged mice where TGF-β receptors significantly decreased with ageing [23]. However, contradictory reports with respect to amounts of TGF-β1 expressed by IVD NP cells creates uncertainty with respect to the role(s) played by this vital growth factor [29]. Therefore, it may be that not all cases of degenerative disc disease behave in the same way with respect to TGF-β1-related signaling. Possibly there is an ideal concentration range of growth factors expressed by IVD NP cells that can lead to further degeneration or attempts at repair. Furthermore, published accounts of TGF-β1 levels and DDD have used different methods of detection and analysis that leads to confusion with respect to the role played by TGF-β1 in homeostatic regulation of the IVD or in possible repair strategies.

## 9. Conclusions

Taken together, the body of work that has accumulated since the seminal paper by Aguiar et al. in 1999 has conclusively demonstrated that notochordal cells secrete vital instructional cues that aid in the homeostatic regulation of the IVD NP. These instructional cues may provide the essential elements of a novel, molecular therapy to treat degenerative disc disease, the ailment responsible for the highest number of years lost to disability worldwide [58,59].

## Data Availability

All data cited within this manuscript are available as published scientific manuscripts and are freely available.

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
