# Peer review of "Current Status of the Instructional Cues Provided by Notochordal Cells in Novel Disc Repair Strategies"

_ijms, 2021, doi:10.3390/ijms23010427_

Round 1

Reviewer 1 Report

Comments for authors

This is an interesting review article to summarize the current status of the influence of notochordal cells on the disc repairment.

Content comments:

In addition to the introduction section, the content of this article has seven major sub-themes.

  1. In the topic “Anatomical/Physiological Role Played by the Notochord in Disc Development and Homeostatic Regulation of the IVD”, CTGF is an important protein, which is secreted by notochordal cells, plays an early rote in development and formation of the notochord. Although TGF-β and Smad signaling provide evidence to develop or maintain the IVD through CTGF/CCN2, a lack of the relationship of TGF-β or Smad and notochordal cells in this article. Can authors give evidence about the association of TGF-β or Smad and notochordal cells?
  2. In the topic “notochordal cell associated extracellular matrix interaction, authors explored that SLRP is the link of notochordal cells and extracellular matrix of disc. As known, SLRP, as extracellular protein, is a role in IVD. Authors cite two studies to confirm expression of SLRP in notochordal cells. Can authors explore the relationship of SLRP and notochordal cells in detail?
  3. In the topic “Notochordal cell secreted proteins”, authors illustrated CTGF and TGF-β are the notochordal cell-secreted proteins and the precious findings are the same as their previous study (Scientific Reports 7, doi:10.1038/srep45623, 2017).
  4. In the topic “Influence of notochordal cells upon mesenchymal stem cells”, the main information is that notochordal cell conditioned medium can induce stem cells to differentiate to an IVD NP like phenotype. However, there was a lack of mechanism of notochordal cells inducing stem cell to differentiated to specific NP.
  5. The topic “Notochordal Cell-Rich Extracellular Matrix Instructional Cues” is interesting, I hope that the authors can explore more detailed information about the mechanism of NCCM and repair of disc or regulation of disc.
  6. In the topic “Influence of Notochordal Secreted Factors Upon Neural Ingrowth into the Degenerative Disc”, the molecular therapy based upon the notochordal cell secretome is a novel treatment for IVD pain, that is an excellent finding.
  7. In the topic “Dichotomy of IVD NP Cell Secreted Growth Factors Upon IVD Health and Disease”, the positive effect of CTGF + TGF-β1 on the IVD is discussed.

Author Response

We thank the reviewer for insightful and helpful commentary.  We have made appropriate changes to the manuscript (changes highlighted) and have attached specific replies to the reviewer here.

Reviewer 2 Report

The present review article does provide an interesting overview on the instructional cues provided by notochordal cells for potential novel disc repair strategies. The outline and scientific content of the article is sensible yet there are considerable shortcomings as far as the language is concerned.

While the introduction and section 2 of the review article read well, the subsequent sections are at times hard to read. This is mostly because of  long and complicated sentence structures , formatting errors or incomplete sentences:

- long sentences (lines 171-173 : 214-219 : 254-257)
- formatting errors (lines 126 : 131 : 224)
- missing words (lines 214-218 : 231-234).

Section 3, line 116 starts a description on experiments performed by the authors. The way the authors articulate their findings is rather suited for a primary data publication than a review article (e.g. line 120 "Within this study we found..."). Also the lines 126 and 131 contain formatting errors in the references. This is in stark contrast to section 7 line 231 where the self citation is correct.

Author Response

We thank the reviewer for their helpful comments/edits and have attached specific replies to the reviewers comments here (highlighted) as well as within the body of the revised manuscript.

Response to Reviewer Comments:  Current Status of the Instructional Cues Provided by Notochordal Cells in Novel Disc Repair Strategies

Ajay Matta and William Mark Erwin

We gratefully acknowledge the reviewer’s insightful comments regarding our submitted manuscript and provide answers to questions raised by reviewer #2 here.

Reviewer #2:

  1. We are grateful to the reviewer for constructive comments with respect to lengthy sentences, formatting errors and missing words. We have endeavored to correct these problems within the revised manuscript (track changes).

  1. Section 3, line 116 starts a description on experiments performed by the authors. The way the authors articulate their findings is rather suited for a primary data publication than a review article (e.g. line 120 "Within this study we found..."). Also the lines 126 and 131 contain formatting errors in the references. This is in stark contrast to section 7 line 231 where the self citation is correct.

Thank you for the observations.  We have corrected these issues within the revised manuscript (track changes enabled).